# Mitogen-like Cerium-Based Nanoparticles Protect *Schmidtea mediterranea* against Severe Doses of X-rays

**DOI:** 10.3390/ijms24021241

**Published:** 2023-01-08

**Authors:** Kristina O. Filippova, Artem M. Ermakov, Anton L. Popov, Olga N. Ermakova, Artem S. Blagodatsky, Nikita N. Chukavin, Alexander B. Shcherbakov, Alexander E. Baranchikov, Vladimir K. Ivanov

**Affiliations:** 1Institute of Theoretical and Experimental Biophysics, Russian Academy of Sciences, Pushchino 142290, Russia; 2Moscow Region Pedagogical State University, Moscow 141014, Russia; 3Kurnakov Institute of General and Inorganic Chemistry, Russian Academy of Sciences, Moscow 119991, Russia; 4Institute of Microbiology and Virology, National Academy of Sciences of Ukraine, 03680 Kyiv, Ukraine

**Keywords:** cerium oxide nanoparticles, cerium fluoride nanoparticles, planarians, radioprotection, X-ray

## Abstract

Novel radioprotectors are strongly demanded due to their numerous applications in radiobiology and biomedicine, e.g., for facilitating the remedy after cancer radiotherapy. Currently, cerium-containing nanomaterials are regarded as promising inorganic radioprotectors due to their unrivaled antioxidant activity based on their ability to mimic the action of natural redox enzymes like catalase and superoxide dismutase and to neutralize reactive oxygen species (ROS), which are by far the main damaging factors of ionizing radiation. The freshwater planarian flatworms are considered a promising system for testing new radioprotectors, due to the high regenerative potential of these species and an excessive amount of proliferating stem cells (neoblasts) in their bodies. Using planarian *Schmidtea mediterranea*, we tested CeO_2_ nanoparticles, well known for their antioxidant activity, along with much less studied CeF_3_ nanoparticles, for their radioprotective potential. In addition, both CeO_2_ and CeF_3_ nanoparticles improve planarian head blastema regeneration after ionizing irradiation by enhancing blastema growth, increasing the number of mitoses and neoblasts’ survival, and modulating the expression of genes responsible for the proliferation and differentiation of neoblasts. The CeO_2_ nanoparticles’ action stems directly from their redox activity as ROS scavengers, while the CeF_3_ nanoparticles’ action is mediated by overexpression of “wound-induced genes” and neoblast- and stem cell-regulating genes.

## 1. Introduction

When exposed to ionizing radiation, living organisms take the most damage from reactive oxygen species (ROS) and free radicals, which are formed during the radiolysis of water [1,2,3]. The ROS have a high level of redox activity due to the presence of unpaired electrons, which leads to oxidative damage to all components of the cells [4]. An effective way to reduce the damage from ionizing radiation is to inactivate the abovementioned damaging agents. In order to reach this goal, radioprotectors are used—compounds that shield the organism from damage to its molecules, cells, organs, and tissues mainly by inactivating ROS and other damaging agents [5]. Despite significant progress in the development of radioprotective substances for military use, there is still a need for new selective radioprotectors and radiomitigators for medical applications, in particular for radiation therapy [6,7,8,9,10]. By using new functional nanomaterials and various approaches, including molecular systems, it is possible to enhance the damaging effect of ionizing radiation on tumor cells by changing their radiosensitivity [11].

Cerium dioxide nanoparticles have emerged as a completely new type of antioxidant in recent years. It was demonstrated that these nanoparticles possess exceptional biological activity, which is supposed to be based on their redox potential in biological environments [12,13,14,15,16]. In addition, CeO_2_ nanoparticles demonstrate activities similar to a range of natural redox enzymes (oxidoreductases), including superoxide dismutase (SOD) [17] and catalase (CAT) [18], which are able to scavenge detrimental ROS. The enzyme-like activity of CeO_2_ nanoparticles is commonly thought to be due to the Ce^3+^/Ce^4+^ redox cycle [19]. Further, CeO_2_ nanoparticles possess outstanding free radical scavenging activity and excellent biocompatibility, which makes them promising candidates for a novel generation of radioprotectors [20,21,22]. Recently, it was shown that Ce^3+^-containing nanoparticles are also able to mimic natural enzymes. In particular, cerium(III) fluoride nanoparticles performed well as antioxidants to protect cell cultures from the oxidative damage of hydrogen peroxide [23]. It is believed that Ce^4+^ ions are responsible for catalase- and phosphatase-mimicking activities [16,24] while SOD-mimetic activity correlates with Ce^3+^ content [25]. It was demonstrated that CeO_2_-modified poly-L-lactide scaffolds containing various concentrations of Ce^4+^ and Ce^3+^ used as an effective substrate for cell growth showed different effects on cell spreading, migration, and adhesion behavior depending on the content of cerium species in different valence states [26].

Planarians are invertebrate flatworms with unique regenerative abilities. The presence of a large number of stem cells, called neoblasts, allows them to actively replace damaged or dying cells in their body [27]. Due to the high proliferation rate of neoblasts, planarians are extremely sensitive to ionizing radiation [28]. Additionally, high doses of radiation (more than 15 Gy) are detrimental to planarians, as they lead to the death of the entire population of neoblasts and, accordingly, to the impossibility of regeneration [29]. Smaller absorbed doses of ionizing radiation lead to the partial death of neoblasts, while the remaining population of neoblasts is able to completely regenerate the worm’s body [30]. The rapid development of pure phenotypes and high sensitivity to ionizing radiation, combined with new genomic technologies, make planarians a unique experimental model for the discovery of potential radioprotective agents [30]. It is well known that ionizing radiation has a pronounced and measurable effect on planarians that can be easily monitored and quantified, such as the rate of blastema regeneration, the number of neoblasts and their mitotic activity, the degree of DNA damage, and the intracellular ROS level [31]. Recently, CeO_2_ and CeF_3_ nanoparticles were shown to act as inorganic mitogens in regenerating planarians [32]. In the current study, we studied the radioprotective action of 2 different types of cerium-containing nanoparticles, including the molecular mechanisms of their bioactivity and the analysis of key gene expression and signaling pathways using a unique in vivo experimental model, the freshwater flatworm *Schmidtea mediterranea*.

## 2. Results

### 2.1. Cerium-Containing Nanoparticles Have a High Degree of Crystallinity

The stable aqueous sols of highly crystalline CeO_2_ and CeF_3_ nanoparticles were obtained by soft chemistry methods. The selected area electron diffraction patterns of the samples (Figure 1a,b) correspond to the space groups Fm3 ®m and P6_3/mcm for the cubic structure of CeO_2_ and the hexagonal structure of CeF_3_. Both cerium-containing materials are characterized by a high degree of crystallinity of CeO_2_ and CeF_3_ nanoparticles [33,34]. The zeta potentials of cerium(IV) oxide and cerium(III) fluoride sols are −35 mV and +39 mV, respectively, which confirms their high colloidal stability, which is also confirmed by the long storage time (more than 1 year) of the synthesized samples without signs of sedimentation. The hydrodynamic radii of CeO_2_ and CeF_3_ nanoparticles in sols diluted with distilled water were 4.8 ± 2.2 and 30.4 ± 16.2 nm, respectively, indicating a low degree of particle agglomeration. Transmission electron microscopy confirms the ultra-small size (2–4 nm) of CeO_2_ nanoparticles. In turn, according to TEM data, the size of CeF_3_ nanoparticles was 15–25 nm with characteristic faceting, which confirms their high degree of crystallinity.

### 2.2. CeO_2_ and CeF_3_ Nanoparticles Show Pronounced Antioxidant Activity after X-ray Irradiation

The data presented in Figure 2 shows that both cerium(IV) oxide and cerium(III) fluoride nanoparticles are capable of preventing the formation of hydrogen peroxide in water upon X-ray exposure due to their catalase-like properties. The cerium(IV) oxide at a concentration of 10^−7^ M provides a decrease in the concentration of hydrogen peroxide formed after irradiation of the solution, while CeF_3_ nanoparticles do not lead to a decrease in the level of hydrogen peroxide at this concentration. An increase in the concentration of nanoparticles of both cerium(IV) oxide and cerium(III) fluoride to 10^−6^ M provides a notable antioxidant effect, which is expressed in significant (up to 30%) decrease in the concentration of hydrogen peroxide. The nanoparticles of cerium(IV) oxide and cerium(III) fluoride at a concentration of 10^−5^ M reduced the level of hydrogen peroxide to zero values. Further, the existing reports confirm that the decomposition of hydrogen peroxide by CeO_2_ nanoparticles can be well described using the Michaelis-Menten equation, commonly used to describe enzyme-substrate interactions [35]. The rate of hydrogen peroxide decomposition depends directly on the size of nanoparticles, i.e., on the number of available surface sites for binding hydrogen peroxide molecules. At high peroxide concentrations, when almost all Ce^3+^ − Vo − Ce^3+^ sites are involved in peroxide decomposition, the process of slow oxidation of Ce^3+^ → Ce^4+^ transforms into fast redox cycling, and Ce^3+^/Ce^4+^ oscillations are observed [36]. The possible reasons for different cerium(IV) oxide and cerium(III) fluoride catalase-like activities can be explained not only by different ratios of the Ce^3+^ and Ce^4+^ fractions on the nanoparticles’ surfaces, but also by the number of binding sites for hydrogen peroxide, which directly correlates with the nanoparticle size. The catalytic decomposition of hydrogen peroxide is a heterogeneous process that occurs at the interface and depends strongly on a specific surface area. Cerium(IV) oxide nanoparticles have an ultra-small size (2–4 nm), which corresponds to ≈80% of surface cerium atoms, while cerium(III) fluoride nanoparticles have a size of 15–25 nm, which corresponds to ≈10% of surface cerium atoms [37,38]. Thus, the specific surface area of a cerium(IV) oxide nanoparticle is much larger, resulting in higher catalase-like activity.

### 2.3. CeO_2_ and CeF_3_ Nanoparticles Exhibit Radioprotective Properties on a Planarian Model, with CeF_3_ Acting at Nanomolar Concentrations

In order to evaluate the radioprotective effect of CeO_2_ and CeF_3_ on regenerating planarians, the animals were kept for a day in a solution of CeO_2_ and CeF_3_ nanoparticles (10^−4^ M to 10^−11^ M), then the worms were subjected to X-ray radiation (in doses of 10 Gy and 15 Gy) and decapitated with subsequent measurement of the growing blastema area. The irradiated worms without nanoparticle treatment and the non-irradiated worms served as negative and positive controls, respectively. In addition, both CeO_2_ and CeF_3_ nanoparticles promoted an accelerated growth of the blastema when compared to untreated animals (Figure 3 and Figure 4). In CeO_2_, the most pronounced effect was observed on the 3rd day of regeneration after irradiation of animals at a dose of 10 Gy at a concentration of 10*^−^*^5^ M; the level of protection was 96% higher than the control. At a dose of 15 Gy, the level of protection was 74% for concentrations of 10*^−^*^4^ and 10*^−^*^5^ M (Figure 3).

It was found that cerium fluoride nanoparticles act as a more effective radioprotector than cerium dioxide ones since a similar effect of radioprotection was achieved in their case even at a nanomolar concentration (10*^−^*^9^ M) (Figure 4). However, this radioprotective effect was not pronounced on the 6th day of regeneration. At higher doses of radiation (15 Gy), the greatest effect was observed at all periods of regeneration time (3–6 days) (Figure 4). Significantly, the cerium(III) chloride solution taken as a control at a similar concentration (10^−6^ M) did not show any radioprotective activity (Appendix A), which confirms our hypothesis that only cerium-containing nanoparticles are effective antioxidants and ROS scavengers, but not free cerium ions. Moreover, fluoride ions by themselves are known to take part in the development of oxidative stress [39,40] and suppression of antioxidant enzyme activity processes [41]. The fluoride ions were found to negatively impact nervous system activity and development in planarians, including regeneration processes [42].

### 2.4. CeO_2_ and CeF_3_ Nanoparticles Help Preserve Mitotic Activity after Irradiation

In order to evaluate the effect of cerium dioxide and cerium fluoride nanoparticles on the mitotic activity of regenerating planarian neoblasts, changes in the number of mitotic cells were assessed using immunohistochemical studies (Figure 5). In planarians pretreated with CeO_2_ or CeF_3_ nanoparticles, before irradiation with X-rays at doses of 10 and 15 Gy, mitotic activity remains at levels of up to 70% for cerium dioxide and up to 30% for cerium fluoride. Even at a higher dose of 15 Gy, the protective action of nanoparticles can be clearly observed, differing significantly from the negative control. Since neoblasts are the cells responsible for mitotic activity in regenerating planarian blastema, pretreatment with nanoparticles presumably leads to the survival of a significant portion of neoblasts after irradiation, thus confirming their radioprotective property.

### 2.5. Quantitative Analysis of Cerium Content in Planarians

In quantifying the content of cerium in planarians upon long-term (48 h) incubation, inductively coupled plasma mass spectrometry was used. It was found that when cerium oxide and cerium fluoride NPs are added at a concentration of 10^−4^ M, planarians contain a nanomolar concentration of cerium on the second day of incubation (Table 1). The ICP analysis allows for estimating the number of nanoparticles internalized by a planarian. Given that the introduction of a high concentration of nanoparticles (10^−4^ M) ensures that only nanomolar concentrations remain on the second day, it can be argued that even nanomolar concentrations of cerium-containing nanoparticles can effectively act as ROS scavengers and protect the body of planarians from the negative effects of ionizing radiation.

### 2.6. Expression Analysis of Stem Cell Marker Genes Reveals High Stem Cell Stimulatory Activity by CeF_3_

The RT-PCR analysis of the expression of stem cell marker genes in regenerating planarians was performed to study the effect of CeO_2_ and CeF_3_ nanoparticles on the neoblasts (Figure 6). The study showed that, for CeO_2 _, on the third day after decapitation, significant overexpression of two marker genes is observed. One of them was the gene from the sigma class Smed-soxP-1 (which gives rise to the epidermal layer) [43], and the second was from the class of gamma neoblasts, gata456 and prox-1 (which are necessary for the differentiation of progenitors into intestinal cells and for the survival of these differentiated cells, confirming a key role of the gene in the regeneration process and maintenance of the intestine) [44].

On the 10th day after irradiation, the mRNA transcription of neoblast marker genes is noticeably higher in the CeO_2_ group that was exposed to irradiation compared to the 3rd day after irradiation, which indicates the restoration of the stem cell population. In particular, the expression of the Smed-gata456 and prox-1 genes remained elevated and increased in the genes of the zeta class: Smed-soxB-1 and pbx-1, which form the cells of the epidermal layer.

On the 10th day after the treatment of planarians with CeF_3_ nanoparticles, an increased level of expression of almost all the studied genes was observed, allowing us to conclude that cerium fluoride nanoparticles have a more pronounced radioprotective effect compared to cerium dioxide and are highly beneficial for the survival of neoblasts after X-ray irradiation.

Smed-nlk-1 (Nemo-like kinase), Smed-armc1 (Armadillo repeat-containing 1), as well as Smed-fgfr-1 and Smed-fgfr-4 (fibroblast growth factor receptors), are four neoblast-expressed genes encoding proteins that are very similar in structure to signal transduction proteins [43]. The zeta-class fgfr-1 is involved in the signal systems controlling differentiation/growth/migration of stem cells during planarian regeneration [45]. Ogawa et al. suggested that the loss of regenerative activity in X-ray-irradiated planarians, Dugesia japonica, is caused by the disappearance of fgfr-1-expressing cells in the mesenchymal space. In our study, the reduced expression of genes encoding fgfr-1 in X-ray-irradiated planarians *Schmidtea mediterranea* can be leveled to the intact animal ones by both cerium(IV) oxide and cerium(III) fluoride nanoparticles, with cerous fluoride having a faster effect. Moreover, 10 days after irradiation, the CeF_3_-treated animals overexpress the sigma-class Smed-fgfr-4 and Smed-nlk-1 genes. These genes are involved in the early wound response, namely «the primary class» [46].

## 3. Discussion

The destructive effect of ionizing radiation on the cell structure is associated with two main factors: direct damage to DNA through the action of radiation track and indirect damage through the generation of reactive oxygen species (ROS) and free radicals as a result of water radiolysis. The products of water radiolysis (hydroxyl radicals, peroxide, nitroxyl radicals, etc.) have the greatest damaging effect on cellular structures. They are able to oxidize not only nucleic acids but also proteins, invoking crosslinks in them, as well as lipids, initiating lipid peroxidation [47]. A complex analysis was performed of the biological activity of two types of cerium-containing nanoparticles (CeO_2_ or CeF_3_) on planarian regeneration after X-ray irradiation. The study revealed a dose-dependent and regeneration-stimulating effect of both types of nanoparticles. The suggested mechanisms of the radioprotective effects are a significant decrease in intracellular ROS after irradiation, overexpression of wound repair genes, and induction of neoblast mitotic activity. The regenerative capacity of planarians depends on the population of their adult pluripotent stem cells. The effects of radiation-induced inhibition of planarian regeneration are associated with the partial or complete death of planarian stem cells after X-ray irradiation. Our results show that the neoblasts preserved in the worm body after X-ray irradiation in the presence of cerium-based nanoparticles give rise to a new population of neoblasts and, thus, ensure the regeneration of the planarian body. The molecular mechanisms of proliferation, migration, and differentiation of neoblasts have previously been thoroughly studied using various modern methods, including RNA interference [48]. This makes it possible to identify the influence of external factors, including ionizing radiation, on the processes of regeneration and vital activity of the planaria. The observed radioprotective effects can also be explained by a decrease in the number of ionization products due to their capture and scavenging by nanoparticles. As a result, neoblasts remain partially or completely protected from the effects of ionizing radiation and continue to drive the process of regeneration in planarians. Any impact on a biological system, especially a non-specialized one, implies a biological response in the form of simultaneous stimulation of many independent cellular functions, each of which, in turn, is regulated by a variety of interacting receptors and signaling pathways. The triggering of such complex pathways ultimately causes a metabolically complex and coordinated response at all levels of the organization of the biological system.

Previously, we showed that CeO_2_ nanoparticles are able to effectively neutralize the radiolysis products after exposure to X-rays, reducing the concentration of hydrogen peroxide to almost zero at a nanoparticle concentration of 10**^−^**^5^ M [49]. The antioxidant and radioprotective properties of CeO_2_ nanoparticles were demonstrated in vitro using mouse fibroblast cell culture and in vivo on SHK laboratory mice. It should be noted that CeO_2_ nanoparticles were effective both as a radioprotector (administration before irradiation) and as a radiomitigator (administration after irradiation), ensuring the survival of more than 50% of the experimental group after total irradiation at a lethal dose. Our data show that CeO_2_ nanoparticles effectively penetrate into the body of the worm and provide radioprotection from high doses (10 and 15 Gy) of X-rays, modulate the expression of key blastema regeneration genes, and also maintain the pool of viable stem cells (neoblasts) that provide subsequent blastema growth. It should be noted that after incubation of planarians with nanoparticles, only nanomolar concentrations of cerium (3.43 × 10^−9^ M per 1 planarian) are found in them, despite the initially high concentrations of the introduced nanoparticles (10^−4^ M) (Table 1). Given that we detect nano- and picomolar concentrations of cerium in the whole body of the planarian, detection of cerium directly in the blastema part is impossible due to the ultra-low content and limited sensitivity of the device. At the same time, according to the data on the rate of blastema regeneration (Figure 3), the nanomolar concentrations of CeO_2_ nanoparticles showed a statistically significant increase in the rate of regeneration on the 6th day after irradiation. The large concentrations of CeO_2_ nanoparticles did not demonstrate statistically significant differences in the efficiency of regeneration. It is known that an increase in the concentration of CeO_2_ nanoparticles can lead to their aggregation, which may be a limitation for their effective penetration into the body of planarians. Thus, we can indirectly conclude that not all CeO_2_ nanoparticles will directly penetrate into planarian cells. At the same time, despite the fact that only nanomolar concentrations of nanoparticles remain in planarians, such pretreatment of animals provides high radioprotective efficiency due to their unique antioxidant activity. The similar activity of CeO_2_ nanoparticles was previously demonstrated in various experimental models in vitro. Zal et al. found that cerium oxide nanoparticles reduce the percentage of micronuclei induced by irradiation in lymphocytes by up to 73% [50]. The cell pretreatment significantly reduced the incidence of IL-1β levels as well as the number of apoptotic and necrotic lymphocytes. Goushbolagh et al. showed that CeO_2_ nanoparticles exhibit pronounced radioprotective properties against normal human lung cells but do not protect cancer cells of the line MCF-7 [51]. Shinpaugh et al. demonstrated the potential of CeO_2_ nanoparticles as an effective radioprotector/radiosensitizer in proton therapy [52]. It was shown that pretreatment of normal epithelial cells of the mammary gland with CeO_2_ nanoparticles and their subsequent irradiation with protons with an energy of 3.0 MeV at a dose of 2.8 Gy provided their protection, reducing the proportion of damaged cell nuclei. The direction of the biological action of CeO_2_ nanoparticles can be changed by using different irradiation energies. For example, Briggs et al. showed the multidirectional effects of CeO_2_ nanoparticles in cells irradiated with different energies (10 MV or 150 kVp) [53]. The analysis of the survival curve of radioresistant 9L cells at 150 kVp irradiation indicates a change in the quality of radiation, which becomes more lethal for irradiated cells exposed to CeO_2_ nanoparticles. The authors attribute this change in efficiency to an increase in the generation of Auger electrons with high linear energy transfer at 150 kVp. This selectivity of action makes CeO_2_ nanoparticles quite promising as a theranostic agent capable of acting as a radioprotector/radiosensitizer depending on the irradiation scheme, radiation source, and irradiation energy, which makes it possible to control their biological activity.

The cerium oxide nanoparticles are currently regarded as promising antioxidants and antiproliferative agents with outstanding potential. They improve muscle, gastrointestinal, and retinal function in animal experiments [29,54]. They may find especially important applications for cancer treatment [55]. Interestingly, their high redox activity, which makes them useful as radioprotectors, is also involved in the mechanism of their anticancer action. While redox switching is increasingly recognized as playing an important role in ROS-dependent cancer therapy, ROS-independent cytotoxicity mechanisms such as Ce^4+^ dissolution and autophagy are also becoming important. Despite the fact that pro-oxidant cancer therapy is the most intensively studied, antioxidant activity, capable of protecting healthy tissues surrounding the tumor, also plays an important role by reducing side effects [56].

The biological effects of cerium fluoride nanoparticles were studied much less until recent times. Previously, we have shown that cerium fluoride nanoparticles effectively protect organic molecules and cells from oxidative stress induced by hydrogen peroxide [23]. The doping of CeF_3_ nanoparticles with terbium ions allowed us to provide visualization of nanoparticles [39]. It was also demonstrated the mitogenic activity of cerium fluoride nanoparticles in nanomolar concentrations on regenerating planarians [30]. In expanding on these findings, we compared the regenerative and radioprotective potential of Ce^4+^-containing nanomaterials (CeO_2_ nanoparticles) and Ce^3+^-containing nanomaterials (CeF_3_ nanoparticles) and came to intriguing results. Although both types of nanoparticles possess radioprotective and regenerative properties, the effect of CeO_2_ nanoparticles on planarian regeneration seems to be mediated through their high antioxidative and/or ROS-scavenging abilities. CeF_3_ nanoparticles, on the other hand, seem to act via another pathway: by affecting gene expression and/or signaling pathways. The importance of genetic mechanisms for controlling the regeneration of various planarian tissues is well known [57]. Additionally, by activating an orchestra of wound-induced and other pro-regenerative and antioxidant genes, cerium-based nanoparticles are a more sophisticated and newly discovered class of inorganic bioregulators with radioprotective potential.

As part of this work, we demonstrated for the first time the radioprotective properties of cerium fluoride on the experimental model in vivo. The cerium fluoride nanoparticles, as well as cerium oxide nanoparticles, penetrate planarian cells (8.01 × 10^−9^ M per 1 planarian) and provide radioprotective activity after irradiation at the maximum radiation dose (15 Gy). It was shown that, starting from the 3rd day of observation, regeneration in planarians treated with CeF_3_ nanoparticles was statistically significantly higher compared to the untreated, irradiated control. Such a stimulating effect persisted throughout the observation period up to day 6, which confirms the antioxidant and mitogenic properties of cerium-containing nanoparticles. The modulation of expression levels of key genes in the presence of CeF_3_ nanoparticles on the 10th day of observation after their irradiation demonstrates the long-term bioactivity of nanoparticles on the transcriptional profile of neoblasts. Further, both CeO_2_ and CeF_3_ nanoparticles are proven to be effective radioprotectors, in the range of micromolar or even nanomolar concentrations on the planarian model. The outstanding and previously unknown effect of CeF_3_ particles is probably caused by their ability to stimulate genes responsible for planarian neoblast proliferation. This result seems to have very high biomedical prospects, taking into account the low toxicity of cerium fluoride: the single-dose acute oral LD50 is greater than 5.0 g/kg (Sprague-Dawley rats), CeF_3_ is not considered to be a skin or eye irritant (New Zealand Albino rabbits) [58]. Meanwhile, the molecular mechanisms and long-term effects of cerium-containing nanoparticles require further research.

## 4. Materials and Methods

### 4.1. Synthesis of Cerium Nanoparticles

The CeO_2_ nanoparticles were synthesized by the hydrothermal microwave method in accordance with the previously described procedure [33]. In addition, the CeF_3_ nanoparticles were synthesized by precipitation in alcoholic media with the previously described procedure [23]. The UV-visible absorption spectra of CeO_2_ and CeF_3_ nanoparticles were measured in standard quartz cuvettes using a UV5 Nano spectrophotometer (METTLER TOLEDO, Zurich, Switzerland). The transmission electron microscopy and electron diffraction (SAED) analysis were performed using a Leo 912 AB Omega electron microscope. Hydrodynamic radii were measured by dynamic light scattering on a N5 submicron particle size analyzer (Beckman Coulter, Pasadena, CA, USA).

### 4.2. Experimental Object

In our work, we used an asexual laboratory strain of freshwater flatworm, *Schmidtea mediterranea* (*Turbellaria*, *Platyhelminthes*). The animals were kept at room temperature in darkened glass aquariums containing artificial freshwater flatworm water (a mixture of tap and distilled water, 2:1 vol). The freshwater flatworms were fed twice a week with mosquito larvae (*Chironomidae*). In addition, before the experiment, flatworms were starved for one week. Animals with a body length of about 10–12 mm were selected for the experiments. The amputation of 1/5 of the anterior part of the flatworm body with the head nervous ganglion (i.e., decapitation) was performed. Further, before the operation, flatworms were immobilized and anaesthetized on the cooling table. The operations were performed under a Carl Zeiss Stemi 2000 dissecting microscope, using a thin eye scalpel. One day before decapitation and before X-ray irradiation (if required by the experiment design), different concentrations of CeO_2_ or CeF_3_ nanoparticles were added. After amputation of 1/5 of the planarian body part containing the head ganglion, regeneration of the severed part of the body was observed. The number of animals in each group was the same and amounted to 35 pcs.

### 4.3. X-ray Exposure

The X-ray irradiation of the planarians was performed using an X-ray therapeutic machine, RTM-15 174 (Mosrentgen, Russia), at a dose of 10–15 Gy (1 Gy/min), 200 kV voltage, 37.5 cm focal length, and a 20 mA current. For irradiation, animals were placed in Petri dishes (35 mm) on filter paper moistened with water.

### 4.4. Assessment of CeO_2_ and CeF_3_ Nanoparticles’ Antioxidant Activity

To determine the antioxidant activity of CeO_2_ and CeF_3_ nanoparticles, we analyzed the concentration of hydrogen peroxide after X-ray irradiation of CeO_2_ and CeF_3_ nanoparticles by enhanced chemiluminescence using a luminol—4-iodophenol—peroxidase system [49]. The TRIS buffer was used to maintain a constant pH (7.2). The irradiation dose was 5 Gy (1 Gy per min). A liquid scintillation counter, Beta-1 (MedApparatura, Kyiv, Ukraine), operating in the mode for counting single photons (with one photomultiplier and the coincidence scheme disengaged), was used as a highly sensitive chemiluminometer. The high sensitivity of this method allows for the detection of hydrogen peroxide at a concentration of <1 nM. The H_2_O_2_ content was determined using the calibration dependencies of chemiluminescence on the H_2_O_2_ concentration in the solution. The concentration of hydrogen peroxide used for the calibration was determined spectrophotometrically at 240 nm using a molar absorption coefficient of 43.6 M^−1^ × cm^−1^.

### 4.5. Intravital Computer Morphometry

In investigating the growth of the regeneration bud (blastema), computer morphometry was used [32]. The control and experimental groups of freshwater flatworms were photographed with a Carl Zeiss AxioCam MRC camera and a Carl Zeiss Stemi 2000 microscope, 72 h after decapitation. The area of the blastema (s) and the total area of the body (S) were determined using the Plana 4.0 software. The index of regeneration, R = s/S, was used as a quantitative measure of blastema growth. Each R-value was calculated as the mean for 30 animals in either the experimental or control group. Each experimental point was repeated in triplicate. The relative change was calculated as follows:$$\Delta R=\frac{\left(R_{E}-R_{c}\right)\pm \left(\delta_{E}-\delta_{c}\right)}{R_{c}}\times 100\%$$

In addition, R*_E_* is the index of regeneration in the experimental group of flatworms; R*_C_* is the index of regeneration in the control group of flatworms; ΔR is the difference (%) between R*_E_* and R*_C_*; δ*_E_* and δ*_C_* are the standard errors of measurement in the experimental and control groups, respectively. The results presented here are the means from three independent experiments. The standard errors in all the experiments did not exceed 6%.

### 4.6. Whole-Mount Immunocytochemical Study of Planarian Stem Cell Mitotic Activity

In this study, planarians with body lengths of about 4 mm were selected. The number of mitotic cells in the regenerating worms was determined after seven days. The planarians were treated with cerium-containing nanoparticles overnight and fixed in PBS containing 4% formaldehyde and 0.3% Triton X100 for 20 min. In addition, the planarian staining for detecting mitotic cells was performed according to the protocol provided by Newmark and Alvarado [31]. In labeling mitotic cells, a primary antibody was used for phosphorylated histone H3 (Santa Cruz, Dallas, TX, USA) at a 1/1000 dilution. A secondary antibody conjugated to a fluorescent label, CF488A (Biotium, Fremont, CA, USA), was used in a 1/1000 dilution. The phosphorylated H3 histone has long been used as a classical marker of mitotic cells in studies of planarian neoblast mitotic activity [32]. After washing in PBS, the whole-mount preparations were placed in Vectashield Antifade Mounting Medium (Vector Labs, Burlingame, CA, USA) and analyzed using a ZEISS Axiolab 5 Fluorescence Digital Microscope (Carl Zeiss, Jena, Germany). The mitotic cell number and the planarian body area were measured using the Carl Zeiss Axio Image software. The number of mitotic cells per 1 mm^2^ of the planarian body (the mitotic index) was then calculated. The average values of the mitotic indices (i.e., the relationship of the total number of mitotic cells to the body area of each animal) were obtained using 15 animals per experimental group in three experimental repetitions. The specificity of immunocytochemical staining was confirmed using a non-immune serum. All controls were negative and demonstrated the absence of specific and non-specific fluorescent staining in planarian tissues.

### 4.7. Real-Time PCR

Furthermore, after incubation of the animals in the presence of CeO_2_ and CeF_3_ nanoparticles, mRNA from experimental (*n* = 5) and control animals (*n* = 5) was extracted with magnetic particles using an mRNA purification kit (Sileks, Russia). The mRNA concentration was measured using a NanoDrop spectrophotometer (Gene Company, South San Francisco, CA, USA). The reverse transcription was performed using the oligo dT primer according to the protocol provided by the manufacturer (Sileks, Russia). The resulting cDNA was amplified as a real-time PCR template using SybrGreen (Syntol, Russia). The polymerase chain reaction was carried out using a BioRad CFX-96 amplifier (USA). The expression of 46 genes that control the early stages of regeneration and the proliferative activity of neoplasms, divided into four classes (W1, W2, W3, and W4), was measured [31]. In addition, the expression of another 15 key genes involved in regeneration was measured: ζ-class neoblast subpopulations (ancestors of all neoblasts), σ-class (epidermal progenitors), and γ-class (interstitial cell progenitors). The level of gene transcription was normalized according to the average levels of transcription of the housekeeping genes Smed-ef1 and Smed_01699. The genomic DNA contamination was determined from a sample without a reverse transcription stage based on genome-specific primers. Further, gene-specific primers were selected using the Primer Express program (Applied Biosystems, Waltham, MA, USA) (Appendix A). The obtained expression data were analyzed using the online service http://www.qiagen.com (accessed on 25 January 2022), the mayday-2.14 program (Center for Bioinformatics, Tübingen, Germany), and the Genesis program.

### 4.8. Inductively Coupled Plasma Mass Spectrometry (ICP-MS)

The planarians were incubated for two days with CeO_2_ and CeF_3_ nanoparticles (10^−4^ M), then the sample preparation procedure was carried out for analysis by the ICP-MS method. In addition, the planarians were washed three times with double-distilled water and placed in a 30% H_2_O_2_ solution (Sigma Aldrich, St. Louis, MO, USA) under bright light for 16 h for decolorization. After this procedure, concentrated HNO_3_ (Khimmed, Moscow, Russia) was added to the sample, and the samples were placed in a dry oven at a temperature of 150 °C until the solution evaporated. Further, after drying, nitric acid was added to the sample and measured using an Element™ Series HR-ICP-MS analyzer (Thermo Scientific, Waltham, MA, USA).

### 4.9. Statistical Data Processing

The experiments were performed in 3–4 repetitions, with three independent repetitions for each concentration of CeO_2_ and CeF_3_ nanoparticles. The experimental results were compared with those of untreated controls. In addition, the statistical analysis was performed using the methods of variation statistics (ANOVA, Mann–Whitney U test). The means and standard deviations (SD) of the means were determined. The significance of differences between groups was determined using a Student’s *t*-test. The obtained data were processed statistically using the Sigma-Plot 9.11 program (Systat Software Inc., Erkrath, Germany).

### 4.10. Ethical Standards

All procedures performed in this study involving animals were performed in accordance with the ethical standards of the institution at which the studies were conducted.

## Figures and Tables

**Figure 1 ijms-24-01241-f001:**
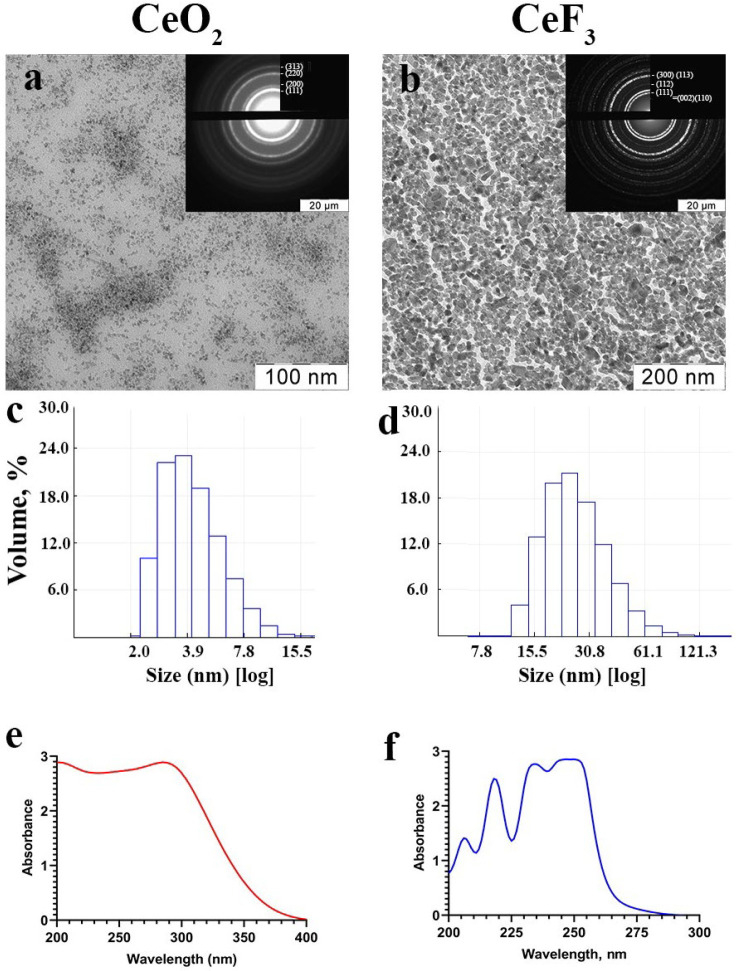
Transmission electron microscopy (**a**,**b**), dynamic light scattering (**c**,**d**) in water, and UV absorption spectra (**e**,**f**) of cerium oxide (CeO_2_) and cerium fluoride (CeF_3_) nanoparticles. Insets in (**a**,**b**) show selected-area electron diffraction data.

**Figure 2 ijms-24-01241-f002:**
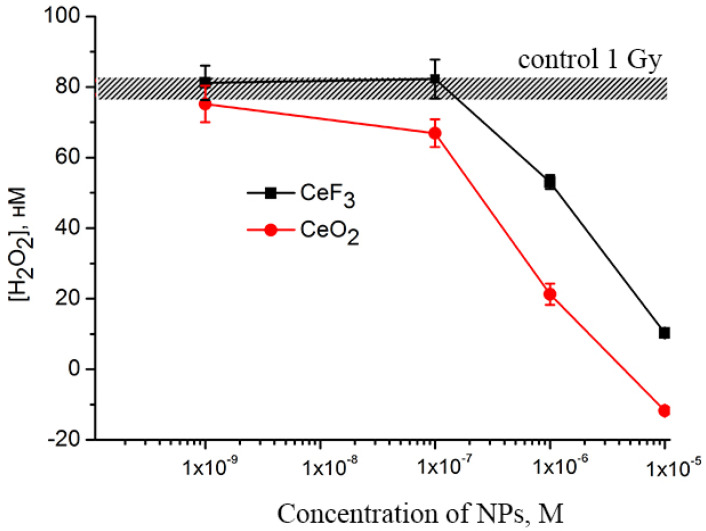
Concentration dependencies of hydrogen peroxide formation induced by X-ray irradiation (1 Gy) in a buffer solution containing CeF_3_ and CeO_2_ NPs. The mean values of three independent experiments and their standard errors are given.

**Figure 3 ijms-24-01241-f003:**
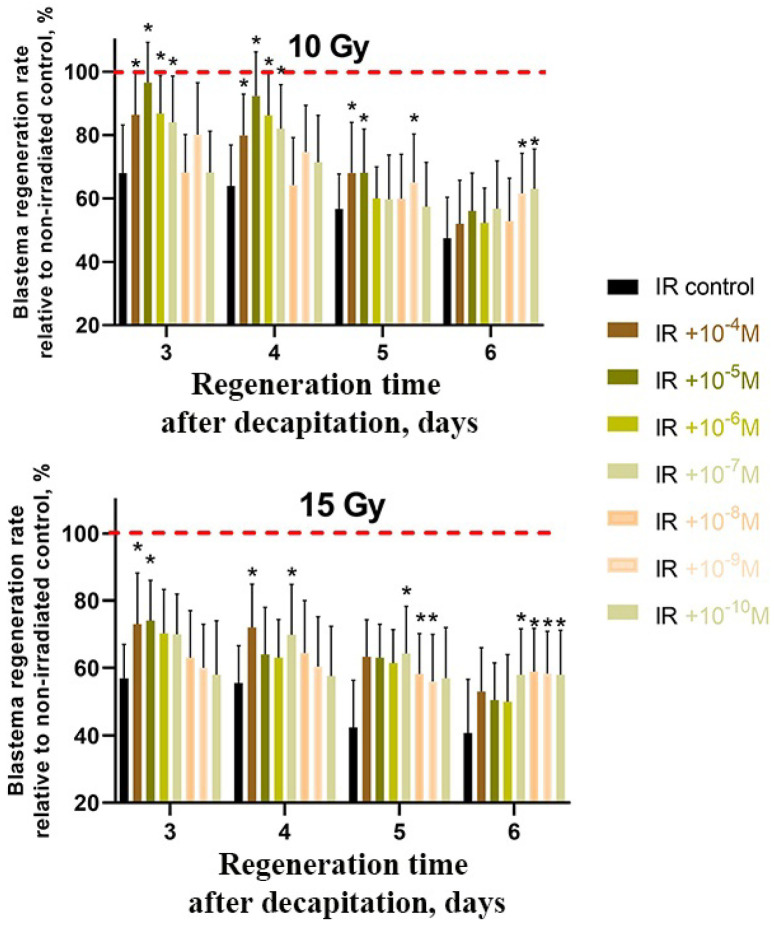
Radioprotective effect of CeO_2_ nanoparticles on regenerating planarian blastemas after X-ray irradiation (10 Gy and 15 Gy). Brown to light-brown bars represent the percentage of blastema regeneration rate; the regeneration rate of non-irradiated animals was taken as 100% (red line). * *p* < 0.05 (difference from the control group). M ± SD, *n* = 90. IR—X-ray irradiation.

**Figure 4 ijms-24-01241-f004:**
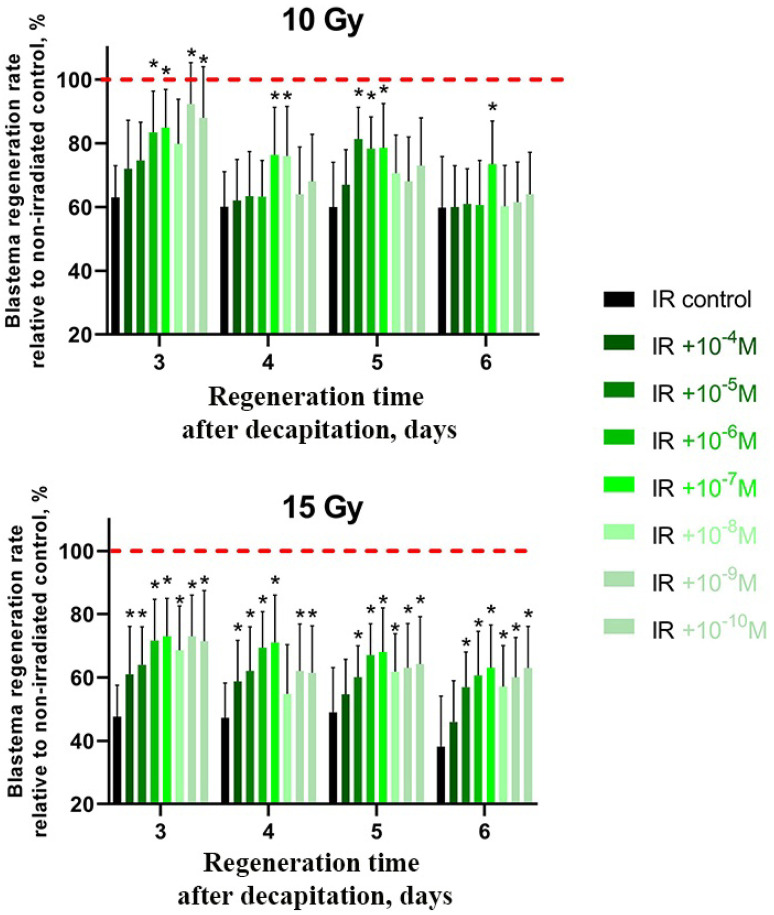
Radioprotective effect of CeF_3_ nanoparticles on regenerating planarian blastema after irradiation with 10 Gy and 15 Gy. Green to light-green bars represent the percentage of blastema regeneration rate; the regeneration rate of non-irradiated animals was taken as 100% (red line). * *p* < 0.05 (difference from the control group). M ± SD, *n* = 90. IR—X-ray irradiation.

**Figure 5 ijms-24-01241-f005:**
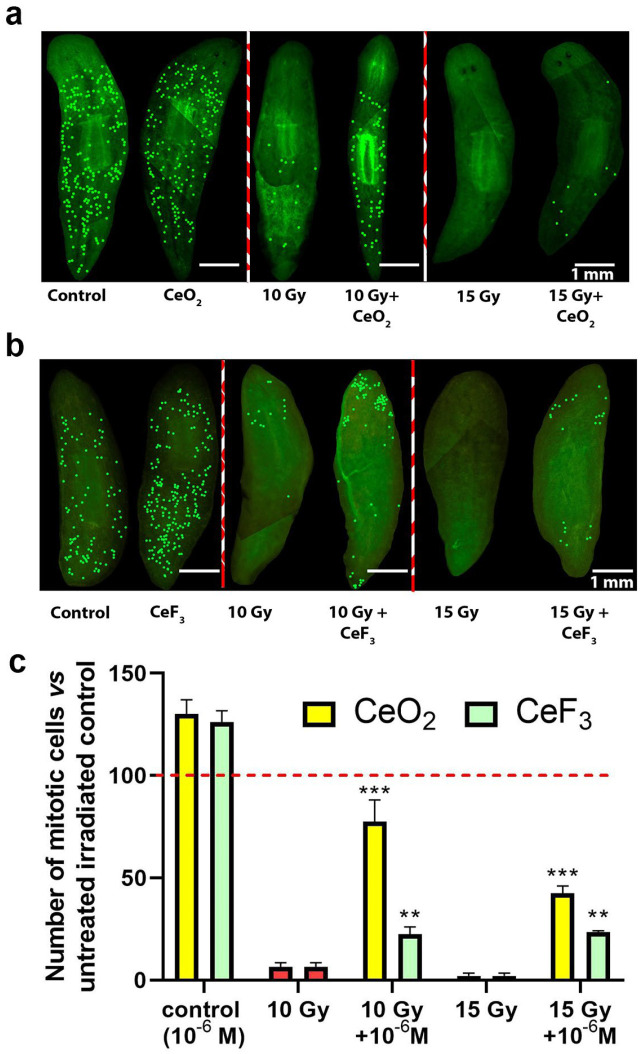
Mitotic activity of regenerating planarians pretreated with CeO_2_ and CeF_3_ nanoparticles and X-ray irradiated. Photographs of immunostained planarians after X-rays irradiation with CeO_2_ (**a**) and cerium fluoride (**b**) nanoparticles. Quantitative analysis of the level of mitotic cells (**c**). This concentration of nanoparticles (10*^−^*^6^ M) was chosen as it gave the most pronounced effect. Significance was estimated by one-way analysis of variance (one-way ANOVA). Data are presented as means. Significant statistical difference ** *p* < 0.001, *** *p* < 0.05 compared to the untreated control (indicated by the red line).

**Figure 6 ijms-24-01241-f006:**
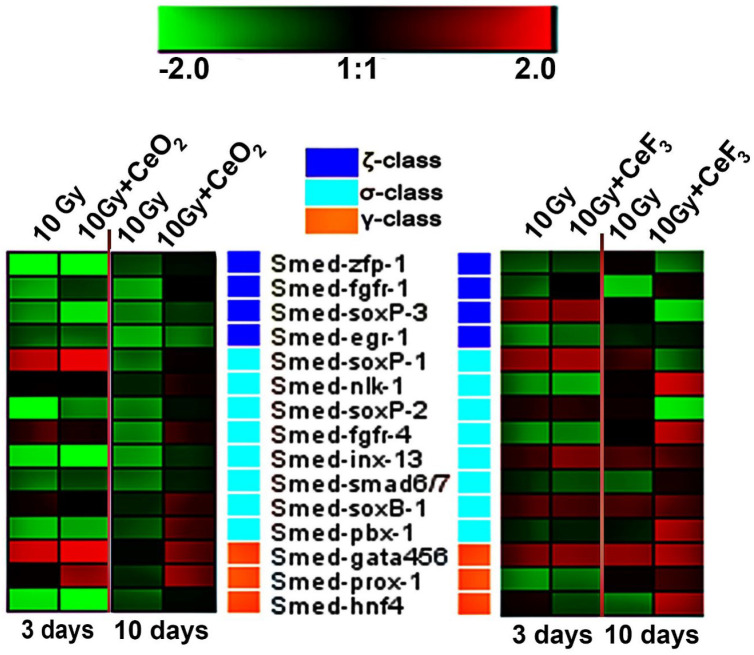
Expression of 3 classes of neoblast marker genes in regenerating planarians on days 3 and 10 after 10 Gy X-ray irradiation. The intensity scale of the standardized expression values ranges from −2 (green: low expression) to +2 (red: high expression), with a 1:1 intensity value (black) representing the control (non-treated). The data in the heat maps are from the non-irradiated control group. A non-irradiated control group without CeO_2_ or CeF_3_ nanoparticle pretreatment was taken as a control.

**Table 1 ijms-24-01241-t001:** The content of cerium species in planarians as determined using the method of mass spectrometry with inductively coupled plasma.

Sample Name	Dissolution Medium	Ce Content Per 1 Planarian, M
CeO_2_ + planarians	HNO_3_ + H_2_O + H_2_O_2_	3.43 × 10^−9^
CeF_3_ + planarians	HNO_3_ + H_2_O + H_2_O_2_	8.01 × 10^−9^
H_2_O + planarians	HNO_3_ + H_2_O + H_2_O_2_	-
HNO_3_ + H_2_O + H_2_O_2_	HNO_3_ + H_2_O + H_2_O_2_	-

## Data Availability

The data presented in this study are available in the article.

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
