# Peer review of "Mitogen-like Cerium-Based Nanoparticles Protect Schmidtea mediterranea against Severe Doses of X-rays"

_ijms, 2023, doi:10.3390/ijms24021241_

Round 1

Reviewer 1 Report

This manuscript reports on the protection exerted by nanoparticles containing cerium oxide or fluoride against X-irradiation in planarian flatworms S. mediterranea, frequently used as a model.

The research group has a good experience with the biological properties of these nanoparticles and in previous publications, they showed, on one side, that cerium nanoparticles stimulate planarian regeneration probably by ROS scavenging and gene activation, and on the other side, that cerium oxide nanoparticles exert a radioprotection against X-rays of due to their antioxidant properties. The present report expands the latter study to the modulation of key genes involved in planarian head regeneration and compares cerium oxide and cerium fluoride nanoparticles.

I have two general comments.

1.     There is a lack of Information on the intracellular distribution of nanoparticles. The authors quote ref 39 but this is of little use for the present biological system.  The paper should be improved with this information since it can be important when discussing the mechanisms of action.

2.     In the Discussion it is not clear how much the protection is related to ROS scavenging or modulation of key genes. Since they are effects at different levels, the authors should discuss if there is any relationship between them. In other words, can ROS scavenging/suppression modulate the key gene expression? (See specific comment for lines 231-33)

Specific comments are:

·         Line 36 - Ref 1. I suggest replacing or adding this ref with more specific or basic paper(s) such as:

-       Cadet et al. Oxidative damage to DNA: Formation, measurement and biochemical features. Mutation Research 2003.

-       von Sonntag C. 2006. Free-radical-induced DNA damage and its repair. A chemical perspective. Berlin: Springer.

·         Line 38 - Ref. 2 should be replaced or added with more specific or basic reference(s). See, e.g. :

-       Azzam et al. Ionizing radiation-induced metabolic oxidative stress and prolonged cell injury. Cancer Lett. 2012 and the refs therein.

·         Lines 121 fig2 and 140 Fig3. What is "Regeneration time"? Days after decapitation? In the Supplementary material, it is indicated as Regeneration duration.

·         Line 167 - “long term incubation”: how long? From the Materials and Methods section, it can be deduced it is one day. Why not give quantitative indications here?

·         Line 174 - It is not clear how this ROS scavenging is manifested or measured. Please give this information in this manuscript or quote some refs.

·         Line 192 - “higher” of what ? Please re-word

·         Line 203 - The patterns at 3 and 10 days after 10 Gy irradiation, without nanoparticles, are quite different when comparing the left and right panels. If this is an indication of experimental variability, it has to be considered when comparing the result of CeO2 vs CeF3.

·         Line 226 - Change “there” to “they”

·         Lines 231-33 - Explain the (possible) relation, if any, between intracellular ROS, wound-induced gene expression, and activation of planarian cells, since this activation could also be driven by modulation of gene expression that, in turn, may be related to the amount of ROS activity.

·         Line 236 - “The results presented here…”: are the authors referring to irradiation in the absence or in the presence of Ce nanoparticles?

·         Line 243 - "also" means in addition to what ?

·         Line 256-57 - The data here presented only show that the worm protected by CeO2 also overexpress some genes involved in the blastema regeneration. These data are only suggestive, not conclusive, that this gene modulation is the cause of protection

·         Line 260 - "only" not clear, perhaps "even" ?

·         Line 283 – I think that “that” after “Interestingly” must be removed. The sentence is not clear.

·         Lines 285-289 – Sentence not clear. “A significant decrease” compared to what ? Briggs et al showed a significant decrease in radioprotection for 150 keV compared to 10 MeV. Please re-word.

·         Line 312 - ref 39 deals with TEM microscopy, not with confocal laser microscopy as said in the text.

·         SUPPLEMENTARY - It is not clear why the authors provide data on cerium chloride, which is not considered or mentioned in the main text. By the way, the legend of the figure is cut out.

Reviewer 2 Report

In this work by Filippova et al. entitled " Mitogen-like cerium-based nanoparticles protect Schmidtea 2 mediterranea against severe doses of X-rays", the authors aim to discuss the role of CeO2 and CeF3 nanoparticles in promoting head blastema regeneration after ionizing radiation.

The work could be of particular interest to develop new substances working as radioprotectors or radiomitigators for radiation therapy.

However, I have some major concerns and there are different points to consider.

1-    Why did the authors choose doses of 10 and 15 Gy? If 10 Gy is a boost dose that can be used for the treatment of metastasis, 15 Gy is unusual.

2-    The authors found that both nanoparticles accelerate blastema regeneration after irradiation. This phenomenon seems to be dependent on nanoparticles concentration in a biphasic way. How do the authors explain this? Furthermore, in line 112 they stated that two controls have been used, one irradiated without nanoparticles (negative), the other non-irradiated with nanoparticles (positive), but the reviewer cannot find the results concerning the positive control except for the concentration of 10-6 M (Figure 4). How do different concentrations of nanoparticles affect blastema functionality? Could high concentration of nanoparticles have a partial toxic effect?

3-    The amount of cerium species in blastema should quantify for all the concentrations of nanoparticles used in the study and the authors should explain how it is possible that the Ce concentration is very low for the highest concentration considered.

4-    Another important point is related to the significant difference in Ce concentration when CeO2 or CeF3 nanoparticles are used.

Minor concerns:

-       Lines 23-25, CeO2 or CeF3 should be CeO2 or CeF3.

-       Figure 4, please, add scale bar and describe in the caption what is the green signal in a-b.

-       Line 283, 150 kVpk should be 150 kVp.

-       Fig. S1 seems to be cropped.
